# Research on Penetration Loss of D-Band Millimeter Wave for Typical Materials

**DOI:** 10.3390/s22197666

**Published:** 2022-10-09

**Authors:** Xinyi Wang, Weiping Li, Mingxu Wang, Chengzhen Bian, Yi Wei, Wen Zhou

**Affiliations:** Key Laboratory for Information Science of Electromagnetic Waves, Department of Communication Science and Engineering, Fudan University, Shanghai 200433, China

**Keywords:** millimeter wave, penetration loss, attenuation characteristics, channel model

## Abstract

The millimeter-wave frequency band provides abundant frequency resources for the development of beyond 5th generation mobile network (B5G) mobile communication, and its relative bandwidth of 1% can provide a gigabit-level communication bandwidth. In particular, the D-band (110–170 GHz) has received much attention, due to its large available bandwidth. However, certain bands in the D-band are easily blocked by obstacles and lack penetration. In this paper, D-band millimeter-wave penetration losses of typical materials, such as vegetation, planks, glass, and slate, are investigated theoretically and experimentally. The comparative analysis between our experimental results and theoretical predictions shows that D-band waves find it difficult to penetrate thick materials, making it difficult for 5G millimeter waves to cover indoors from outdoor macro stations. The future B5G mobile communication also requires significant measurement work on different frequencies and different scenarios.

## 1. Introduction

The rapid growth of mobile data and the use of smartphones have created an unprecedented challenge for wireless service providers to overcome the global bandwidth shortage [1,2]. To address this challenge, there is growing interest in cellular systems in the 30 to 300 GHz millimeter waveband, which has a much wider bandwidth available than today’s cellular networks [3,4]. Some high frequency bands (mm-band) were previously used for satellite communications, long-range point-to-point communications, military communications, and LMDS (28 GHz), but short wavelengths make it impossible for waves to bypass, or have quasi-optical propagation characteristics [5], which means that the high frequency bands do not have the rich scattering characteristics of the sub-6 GHz band [6,7,8,9,10]. Under line-of-sight conditions, the received signal energy is concentrated on the line-of-sight and a few low-order reflection paths. Under the condition of non-light of sight, signal propagation mainly relies on reflection and bypass, resulting in sparse channels in space and time, and the occlusion of people or objects will lead to large signal fading. High-band channels have many characteristics that are significantly different from sub-6 GHz cellular mobile channels [11,12,13,14]. The development of new 5G systems that can operate in higher frequency bands requires accurate propagation models for these frequency bands. Industry trends at home and abroad show that 5G mmWave is the next stage of 5G development, but it will require a significant amount of time and R&D costs to address the propagation characteristics of mmWave, before it can be deployed as a more general wireless network solution.

Millimeter waves propagate in space as directional waves, have good directivity, are easily blocked by obstacles, and lack penetrating power. Channel measurement and modeling work has also been carried out for high frequency bands, for example, Aalto University in Finland has carried out measurement activities in 15, 28, 60 GHz and E-bands (81–86 GHz) based on a 60 GHz VNA detection system and completed multi-point conference room measurements to obtain the extended SV channel model [15]. Using the VNA measurement system, Ericsson participated in the METIS, mmMAGIC and 5GCM projects and completed the following several measurements: (1) in indoor 60 GHz transmission human blocking experiments [16], it was found that the human blocking loss can also be as high as 10–20 dB; (2) with indoor multi-frequency medium and long-range path loss measurements, it was observed that the bypass is the main path of millimeter-wave indoor non-light of sight transmission [16,17]; (3) multi-frequency under NLOS conditions in urban blocks. The measurement found that the signal path loss does not depend on the frequency very much, and is lower than the result of the knife-edge bypass, indicating that the signal in the case of outdoor NLOS mainly comes from other reflection paths [17]. (4) The final measurement was multi-band measured wall penetration loss [18]. In the 5GCM project, Nokia and Aalborg University in Denmark collaborated on path loss measurements at 10 GHz and 18 GHz [18]. The mm MAGIC project has other measurement activities [17], such as (1) French Telecom Orange Lab (Belfort) completed multi-frequency O2I measurements to observe penetration loss as a function of frequency; (2) French CEA-LETI completed 83.5 GHz and other indoor propagation channel measurements.

In April 2019, Verizon, the largest mobile operator in the United States, launched 5G mobile services in the 28 GHz band in Chicago and Minneapolis. For indoor coverage, Verizon’s 5G mmWave signal is nearly unreachable. After penetrating the concrete wall, the 5G download rate dropped sharply from 600 Mbit/s to 41.5 Mbit/s, while the 4G downlink rate at 1900 MHz did not change much, due to the severe penetration loss of 5G mmWave. Clearly, the wave penetration loss of typical building materials should be studied in light of the need to improve future 5G mmWave indoor coverage. However, the existing millimeter-wave research mainly focuses on the millimeter-wave frequency band below 100 GHz, and the 100–300 GHz millimeter-wave still needs to be developed. D-band (110~170 GHz) electromagnetic waves (EMW), with a frequency range of 110~170 GHz, are located in the cross-band of millimeter waves (30~300 GHz) and terahertz (THz) waves (100~10,000 GHz). The atmospheric window frequency bandwidth of the D-band millimeter wave is about 26 GHz, and its center frequency is about 140 GHz, and the propagation loss in the air is smaller than THz band. Compared to lower millimeter wave frequencies, D-band electromagnetic wave signals have a wider bandwidth, with narrower beams and shorter wavelengths, resulting in greater transmission capacity and higher resolution. Research on the D-band channel propagation characteristics will be helpful to the research of new technologies in the physical layer of 5.5G and even 6G systems. To this end, this paper studies the penetration characteristics of the D-band (140–160 GHz) to typical materials, such as glass, slate, vegetation, and wood, and finds that the penetration loss of D-band millimeter waves is not very dependent on high frequencies, while the masking loss of vegetation is as high as 10–20 dB. D-band millimeter waves can hardly penetrate 5 cm-thick slate and 2 cm-thick wood, and the penetration loss is positively related to the thickness of the masking material.

mmWave is known to increase the capacity of 5G networks and reduce latency. Wider implementations of high-definition video conferencing, teleoperation, and industrial automation will benefit from the wider bandwidth of the mmWave spectrum, especially those applications that require high precision. 5G mmWave will also enable each automated robot to generate or receive large amounts of data, as well as the high-density deployment of these robots in confined areas. From this point of view, good mmWave indoor coverage is necessary. Some current measurements and modeling efforts are still well underway, and work is expected to be carried out in several areas [9], including the following: current measurements focus on a few hotspots, with additional measurements in other candidate frequency bands; existing models claim that they support high bandwidths, but rely on systems that typically have smaller measurement bandwidths and lower angular resolution, so they will also enhance measurements and data analysis in large bandwidths and large antenna arrays. In addition, the statistical parameters provided by the model are all in the form of large table data lists. If the size scale parameters and frequencies, connection types, including antenna height and environmental parameters, can be established, this requires a rethinking of the modeling method.

## 2. Materials and Methods

The cause of wireless path loss is the radiation diffusion of electromagnetic waves and the channel characteristics in the transmission path, so that the received power is smaller than the transmitted power. The free-space path loss model describes the channel propagation characteristics in an ideal propagation environment. Its expression is given as
(1)$$FSPL\left(d,f\right)=20log_{10}\left(4\pi df/c\right)$$
where*d* is the wireless transmission distance;*f* is the transmission frequency;*c* is the speed of light.

It can be observed from the above formula that the free space path loss is only related to the transmission distance *d* and the transmission frequency *f*. When the transmission distance or transmission frequency doubles, the loss is increased by 6 dB. The free-space propagation model is suitable for the wireless environment with an isotropic propagation medium (such as a vacuum), which does not exist in reality and is an ideal model, but the air medium is similar to an isotropic medium.

Moreover, atmospheric attenuation is closely related to altitude, air pressure, temperature, and water vapor density above the Earth. Figure 1 shows atmospheric absorption for free-space paths at sea level (z = 0 km) and at 10 km above sea level under dry conditions (water vapor density w = 0 g/m^3^) and standard conditions (w = 7.5 g/m^3^).

As shown in Figure 1, it is a graph of signal attenuation per kilometer as a function of frequency. It can be observed that the attenuation of electromagnetic wave signals of different frequencies in water vapor and oxygen is different, and there are absorption peaks at several frequency points in the relationship between the resonant frequencies of water vapor and oxygen, and the D-band is just between the two absorption peaks, ranging from 0.01 dB/km to around 2 dB/km. So, the D-band is suitable for long distances up to 100 m millimeter wave communication.

In the real environment, the path loss is related to the presence or absence of occluders, the type of occluders, the thickness of the occluders, as well as the angle of the sender and receiver. The FSPL model does not reflect the actual propagation characteristics. For obstacles of different materials in the transmission path, two modeling schemes of three rays and four rays are used in Ref. [19]. With the proposed scheme, the transmission model of the D-band millimeter-wave signal with obstacles at a distance of 100 m can be simplified, as shown in Figure 2.

The attenuation model given in Figure 2 is adaptive for the different materials with different thickness. For example, the relative permittivity and permeability of iron are very large, and the D-band millimeter-wave signal will have a very large transmission loss under the shielding of the steel plate. It can be considered that the D-band millimeter-wave signal cannot penetrate the steel plate, so the three-ray diffraction model is applied to the steel plate shielding. For relatively thin insulator materials, such as 5 mm-thick glass and 3 mm-thick wood, their penetration loss is not very large, i.e., between 2 dB and 5 dB, so the four-ray method is used for D-band modeling [20]. Four rays include three edge diffraction paths and one transmission path. Unlike thicker insulating materials, such as 5 cm-thick slate and 1.75 cm-thick wood, D-band mmWave signals also have very large transmission losses. Therefore, we also believe that the D-band mmWave signal is impenetrable in this case, while the three-ray diffraction method is suitable for simulating thicker insulator materials.

The penetration performance of millimeter waves is very poor with increasing frequency. Therefore, in our experiments, the penetration loss of millimeter waves in the 140–160 GHz frequency band is large, and it is also influenced by the dielectric constant, thickness and other parameters of the blocking material. The schematic diagram of signal transmission [21] is shown in Figure 3.

In Figure 3, *P_in_* is the incident signal power, *P_out_* is the transmitted signal power, *P_ref_* is the reflected signal power, and D is the thickness of the barrier. The fading coefficient can be obtained from the transmission attenuation caused by material penetration [21], as follows:(2)$$\zeta =\left|\frac{\sqrt{\varepsilon_{r}}-1}{\sqrt{\varepsilon_{r}}+1}\right|$$
where$\varepsilon_{r}\;$ is the relative permittivity of the barrier materials.

The relationship between *P_in_* and *P_out_* (that is, the transmittivity of penetration selected below) can be expressed as follows:(3)$$\frac{P_{out}}{P_{in}}=\frac{\zeta }{e^{200D-1}}{\left(1-\zeta^{2}\right)}^{2}e^{-2D\chi }$$
(4)$$\chi \approx \frac{\pi tg\delta }{\lambda \sqrt{\varepsilon_{r}}}$$
whereD is the thickness of the obstacles, and its measurement unit is meters;λ is the operating wavelength;tgδ is the tangential loss angle;$\varepsilon_{r}$ is the relative permittivity of the blocking material.

For almost impenetrable barriers, such as slates, the relationship between *P_in_* and *P_out_* can be expressed as follows:(5)$$\frac{P_{out}}{P_{in}}={\left(1-\zeta^{2}\right)}^{2}e^{-2D\chi }$$

Then, the penetration loss can be expressed as follows:(6)$$A\approx -10\mathrm{lg}\left(P_{out}∕P_{in}\right)$$

Conversely, the relationship between *P_in_* and *P_out_* can also be calculated from the penetration loss measured in the experiment. For a more intuitive comparison, we select the transmittance index for comparison.

It should be emphasized that the parameters of relative permittivity and tangent loss angle have been given in Refs [20,22,23,24,25,26,27,28], and we use these parameters directly in the measured D-band mmWave system. The real part of wood permittivity was measured by a quasi-optical Mach–Zahnder interferometer with a backward-wave oscillator and the result was 1.60–1.89 in the D-band [25]. However, some of the parameters were not measured in the case of the D-band, so there is a certain error. The parameters are shown in Table 1.

The experimental setup of the D-band millimeter-wave transmission system is shown in Figure 4. The signal generator (Agilent 83711B, 1–20 GHz) generates an IF signal from 11.4 GHz to 13.5 GHz, which is extended to the D-band (138 GHz to 163.2 GHz) after passing through a six-multiplier and a two-multiplier. D-band signals are transmitted to free space via a standard horn antenna (LB-6-25-A). After passing through the artificially placed shelter, the receiving end is received by the same type of standard horn antenna at the transmitting end. The received signal is first amplified by an electric amplifier with a gain of 30 dB (the specific parameters of LNA are shown in Table 2), and then down-converted to the intermediate frequency (1.2 GHz) in the mixer. At this point, the frequency range of the signal is already within the effective bandwidth of the digital oscilloscope. After that, the IF signal is amplified by an electric amplifier with a gain of 26 dB, and finally captured by a digital oscilloscope (E4407B ESA-E, 9 kHz–26.5 GHz). Therefore, the center frequency and received power of the IF signal are observed. The horn antenna operates from 110 GHz to 170 GHz with a gain of 25 dBi and a half-power beamwidth (HPBW) of 9° in the E-plane and 10° in the H-plane. The sensitivity of the receiver is −56 dBm. The photos of the experimental setup at the transmitter and receiver are shown in Figure 5.

This measuring D-band system is implemented in an indoor environment. It is carried out in the underground garage of Building 2, Jiangwan Campus, Fudan University. The antenna height of the transceivers is 1 m, the distance between them is 100 m, and the angle is horizontally aligned. We used a laser pointer to ensure that the antennas on the Tx-side and Rx-side are aligned. First, no obstructions are placed between the transceivers, and the signal power received without obstruction is measured, so that it can be used as a benchmark. Then, the occluders (the details about the occluders are shown in Table 3) are artificially placed in the space, and the position of the occluders is continuously adjusted to minimize the power loss, so as to complete the calibration. We repeat the measurement of the same parameter 10 times at each frequency point to improve the measurement accuracy, and take the average value as the measurement result at that point. Finally, the received signal power through different obstacles is subtracted from the reference value to obtain the penetration attenuation mainly caused by the obstacles. In order to compare the transmission loss of different materials, the occluders were manually replaced to obtain different transmission loss results.

Figure 6 is a schematic diagram of the relative positions of the occluders and the transceivers in the experiment, including vegetation, a wood board, regular single-layer glass and slate materials. The vegetation includes potted green plants in the laboratory. When the number exceeds one pot, all the green plants are stacked together, as shown in Figure 7.

## 3. Results

Figure 8 and Table 4 show the experimental measurement results of placing occluders of different materials in the 100-m D-band mmWave wireless transmission experiment. It can be observed from the experimental results that the attenuation of the D-band millimeter-wave signal transmission by obstacles is positively correlated with the thickness or number of obstacles. Signal attenuation increases with the thickness and number of obstacles.

For example, in Figure 8a, the penetration loss of one pot of vegetation is about 12 dB, the penetration loss of two pots of vegetation is about 16 dB, and the penetration loss of three pots of vegetation is greater than 21 dB. Due to the irregular distribution of vegetation stems and leaves, the power of the signal transmission path is greatly reduced when the signal transmission path is blocked by vegetation. It is important to emphasize that attenuation varies greatly with the irregular nature of the vegetative medium, which means that penetration losses depend on a wide range of vegetation types, densities, and the actual amount of water contained.

In Figure 8b, the loss of the 0.3 cm-thick board is about 2 dB, the loss of the 0.6 cm-thick board is about 6 dB, and the loss of the 1.75 cm-thick board is greater than 23 dB. Penetration loss may be proportional to the thickness of the occluded board, and D-band mmWave signals cannot penetrate boards thicker than 1.75 cm.

As shown in Figure 8c, the loss of 5 mm-thick glass is about 4 dB, and the loss of 10 mm-thick glass is about 9 dB. Path loss can be positively related to the thickness of the shielding glass. From this result, it can be observed that the penetration loss of the D-band in thin glass is not very large.

The loss of the D-band signal as it passes through the 5 cm-thick slate is about 20 dB in Figure 8d, indicating that the D-band millimeter wave signal can hardly penetrate the 5 cm-thick slate.

However, in the four cases, under the same obstacle, the penetration loss does not change much with the increase in the signal frequency. It can be observed that the penetration loss of the D-band is independent of frequency.

## 4. Discussion

Next, we explore the transmission rate in our conducted D-band transmission system as a function of transmission frequency, thickness and type of blocking material. The theoretical penetration rate is calculated according to the above-modified theoretical model and compared with it, as shown in Figure 9. The experimental measurements are basically consistent with the theoretical values.

Since vegetation is not comparable the other three typical building materials, parameters such as relative permittivity can be queried, so the theoretical transmittance of vegetation is not listed here. In Figure 9a, only the transmittance of vegetation measured experimentally is plotted.

As shown in Figure 9b,c, for both the experimental value and the theoretical value, with the increase in the thickness of the board and glass, the penetration rate will decrease and the loss will increase. However, the experimentally measured transmittance is greater than the theoretical value. This is caused by multipath propagation effects. This phenomenon is related to factors such as the location of the obstacle and the surrounding environment. Since this experiment was conducted in an indoor environment, the received interference included reflections from walls and objects around the experimental site, diffraction from occluders, and scattering from vegetation. Moreover, although the amplitude of the signal that reaches the receiving antenna is small under the occlusion of obstacles, the system error is still relatively large in the case of small signal reception. This error can be improved by reducing the measurement error by increasing the measurement accuracy of the system.

In Figure 9d, for both the experimental measurement and the theoretical analysis, the calculated transmittance of slate is extremely low, and it can be observed that the D-band millimeter wave can hardly penetrate 5 cm-thick slate. According to the general theory, the shielding effect of the material includes the following two parts: reflection shielding and absorption shielding, and the penetration loss effect of slate combines these two factors. Reflective shielding is caused by the impedance mismatch of propagating waves, and absorbing shielding is caused by heat loss from hydrates inside the concrete and steel mesh.

## 5. Conclusions

This paper discusses the penetration loss of D-band millimeter waves when shielded by various materials, such as vegetation, board, glass, and slate, as well as blocking measurement experiments in an indoor environment. The experimental results show that, under the given experimental conditions, the average transmission attenuation of D-band millimeter waves caused by a pot of vegetation is about 12 dB, implying that the receiving antenna receives only about 6.5 percent of the transmit power. As the number of vegetation increases, the attenuation of the D-band millimeter-wave signal increases sharply. In our experiment, when the amount of vegetation is increased to three pots, the receiving end can hardly receive the D-band millimeter-wave signal. For the measurement of the wooden board, the transmittance decreases with the increase in the thickness of the wooden board. Millimeter waves can penetrate thin boards, but when the thickness of the boards exceeds 1 cm, D-band millimeter waves can hardly penetrate obstacles. The average transmission attenuation coefficient of the thin glass shield to the D-band millimeter wave is about 4.4 dB, that is, only about 35% of the transmitted power is received by the receiving antenna. The loss of the D-band signal that passes through 5 cm-thick slate is about 20 dB, indicating that it can barely penetrate 5 cm-thick slate. The experimental measurement values are consistent with the theoretical value in general. The experimental measurement results of various materials show that the influence of occlusions on D-band millimeter-wave transmission cannot be ignored, which has a potential application prospect for estimating the channel attenuation characteristics of 5G or 6G systems with obstructions. In addition, we will explore the transmission loss of more frequency points and more materials in future work.

## Figures and Tables

**Figure 1 sensors-22-07666-f001:**
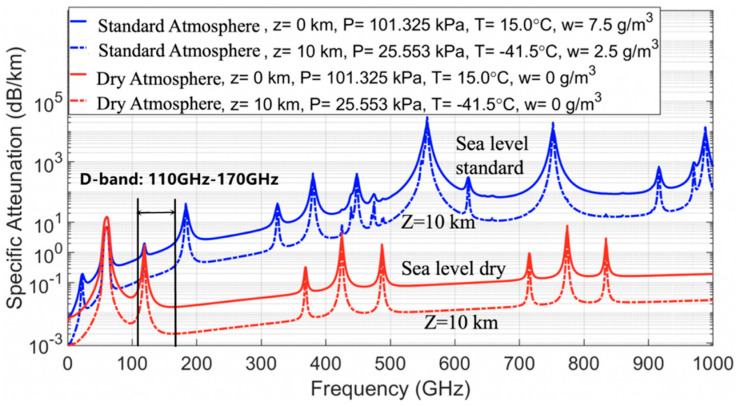
Free-space path atmospheric absorption for dry and standard environments at z = 0 km and z = 10 km.

**Figure 2 sensors-22-07666-f002:**
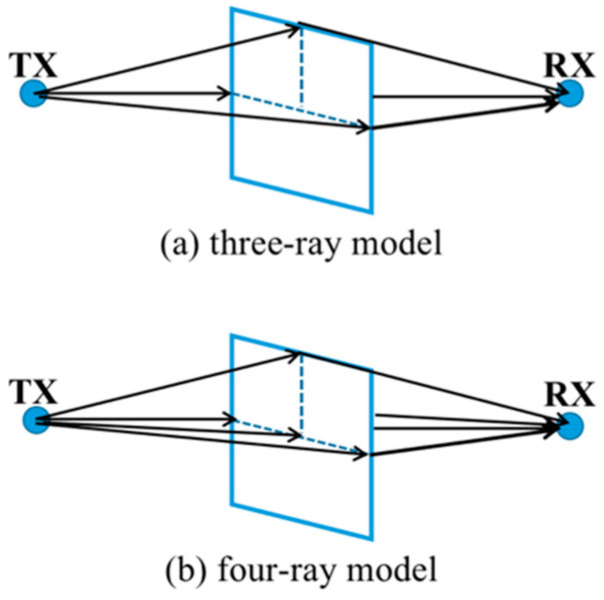
Transmission attenuation model diagrams of D-band millimeter-wave signal passing through obstacles.

**Figure 3 sensors-22-07666-f003:**
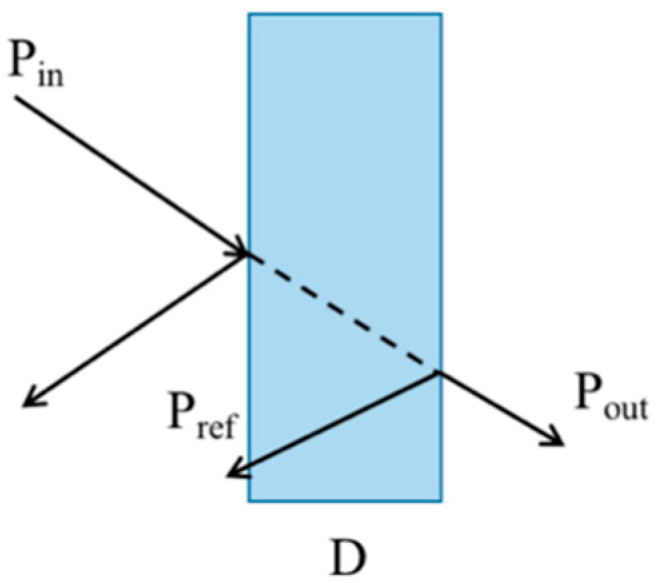
Schematic diagram of D-band millimeter-wave signal transmission.

**Figure 4 sensors-22-07666-f004:**
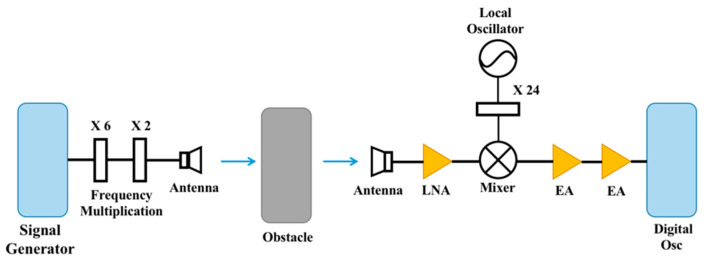
Block diagram of the D-band millimeter-wave transmission experiment system.

**Figure 5 sensors-22-07666-f005:**
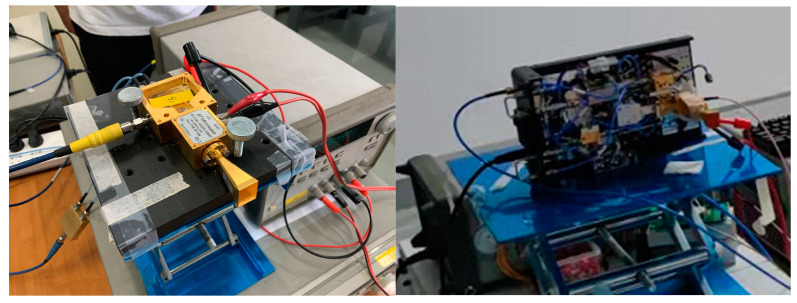
The photos of the experimental setup at the transmitter and receiver.

**Figure 6 sensors-22-07666-f006:**
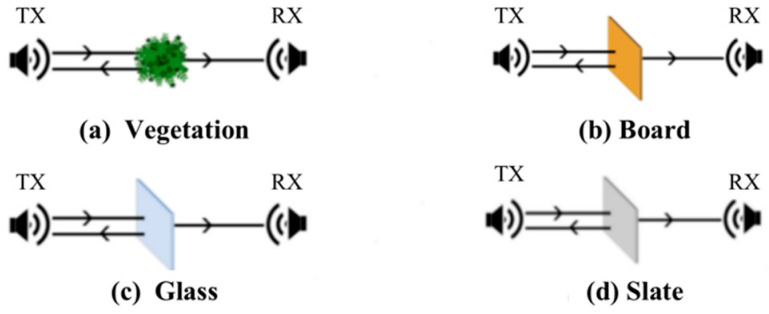
Schematic diagrams of the relative position of the occluders and the transceiver in the experiment.

**Figure 7 sensors-22-07666-f007:**
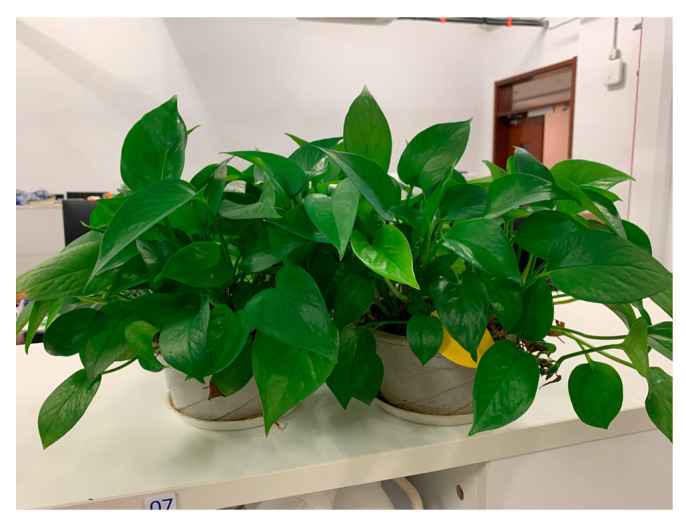
The separation of the pots.

**Figure 8 sensors-22-07666-f008:**
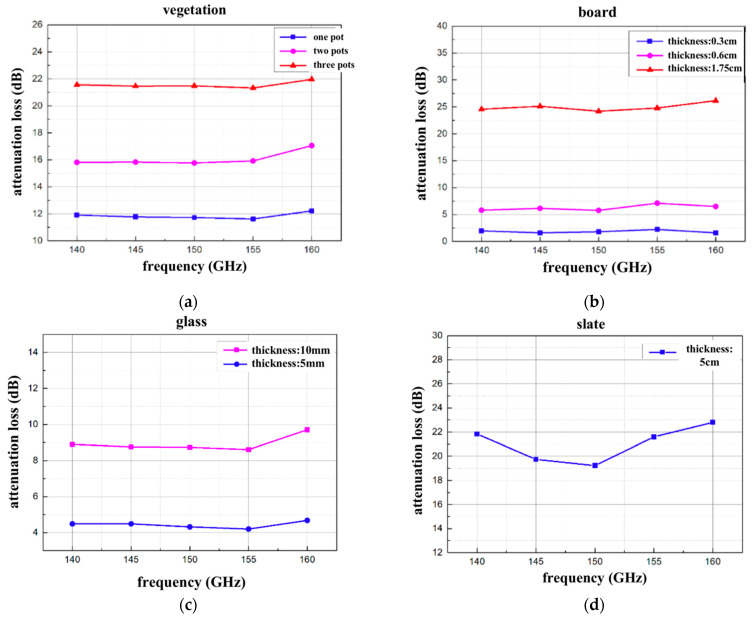
The attenuation loss of D-band millimeter-wave signal through different obstructions measured experimentally. The obstructions are listed as: (**a**) the relationship between penetration loss and frequency under the shielding of different numbers of vegetation; (**b**) the relationship between penetration loss and frequency under the shielding of different thicknesses of wooden boards; (**c**) the relationship between penetration loss and frequency under the shielding of different thicknesses of glass; (**d**) the relationship between penetration loss and frequency under the shielding of 5 cm-thick slate.

**Figure 9 sensors-22-07666-f009:**
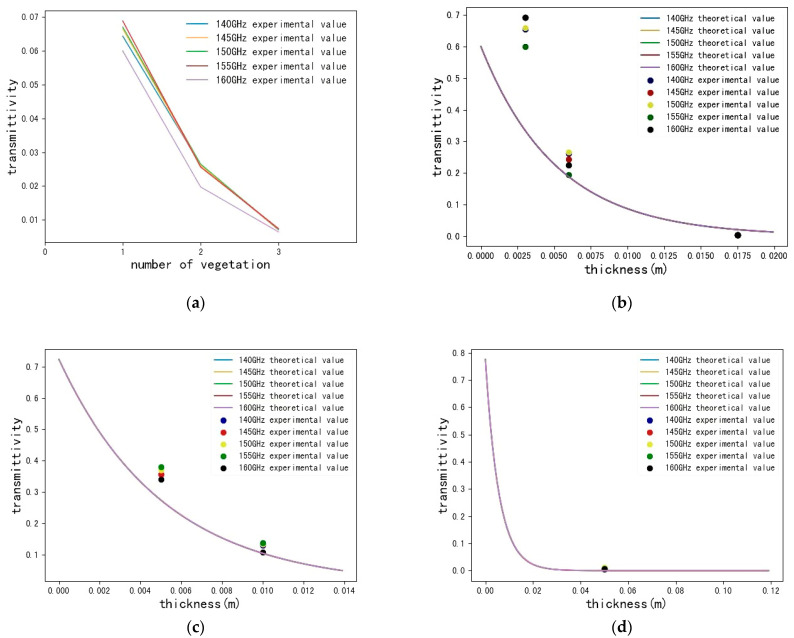
Theoretical and actual transmittance of D-band millimeter-wave signal passing through the obstruction: (**a**) the relationship between the actual transmittance and the amount of vegetation at different frequencies; (**b**) the relationship between the theoretical and actual transmittance and the thickness of board at different frequencies; (**c**) the relationship between the theoretical and actual transmittance and the thickness of glass at different frequencies; (**d**) the relationship between the theoretical and actual transmittance and the thickness of slate at different frequencies.

**Table 1 sensors-22-07666-t001:** Parameters of relative permittivity and tangent loss angle of different materials.

Material	Frequency	$\varepsilon_{r}$	tgδ
Wood	<100 GHz	1.99 [24]	0.0040
	110~170 GHz	1.6~1.89 [25]	0.0040
Plexiglass	143 GHz	2.60 [26]	0.0050
	60~300 GHz	2.581~2.602 [27]	0.0050
Concrete board	1~95.9 GHz	6.2~7 [28]	0.0491

**Table 2 sensors-22-07666-t002:** The parameters of LNA.

Product Type	D-LNA 110-170 30 6
**RF frequency (GHz)**	110-170
**Waveguide designator**	WM-1651 (WR-6.5)
**Gain (typ.) (dB)**	30
**Noise figure (dB)**	6
**P1dB (dBm)**	−3
**Max RF input power (dBm)**	−30
**Part-no.**	03000025

**Table 3 sensors-22-07666-t003:** The details about the occluders.

Material	Size (cm^2^)
Wood board	49.2 × 35.8
Regular glass	88.2 × 42.9
Slate	59.7 × 39.3

**Table 4 sensors-22-07666-t004:** The attenuation loss of D-band millimeter-wave signal through different obstructions measured experimentally.

Material	Thickness (m)/Number	Penetration Loss (dB) 140 GHz	Penetration Loss (dB) 145 GHz	Penetration Loss (dB) 150 GHz	Penetration Loss (dB) 155 GHz	Penetration Loss (dB) 160 GHz
Vegetation	One pot	11.9	11.8	11.8	11.7	12.1
	Two pots	16.0	16.0	15.9	16.1	17.0
	Three pots	21.7	21.5	21.6	21.3	22.0
Glass	0.005	4.5	4.6	4.3	4.1	5.0
	0.01	9.0	8.9	8.8	8.7	10.1
Wood board	0.003	2.5	2.4	2.4	2.5	2.5
	0.006	6.0	6.2	5.9	7.5	6.4
	0.0175	24.9	25.1	23.1	25.1	26.5
Slate	0.05	22.0	19.9	19.2	21.9	22.9

## Data Availability

Not applicable.

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
