# Peer review of "Research on Penetration Loss of D-Band Millimeter Wave for Typical Materials"

_sensors, 2022, doi:10.3390/s22197666_

Round 1
Reviewer 1 Report
There are many experiments and test results for the transmission loss in many materials at the frequency band from 1GHz-500GHz.
Questions and comments:
1. What is B5G? please give the full spelling at the first sight in the paper.
2. In page 3 line 121 around, it is said in the paper that the absorption loss in the entire D-Band is less than 2dB, actually,(1) it should be 2dB/km, (2) the is a big difference for the loss over the D-band from 0.01dB/km to around 2dB/km, about 100 times in fluctuation, please make clear description of the loss in the D-Band.
3. The authors use the four-ray model, then is it possible to use one model with more rays, such as 5 or more?
4. What kind of glass, what kind of board, are used in the experiment?
5. Please tabulate the data in Figure 6 so that it can be known directly by the readers.
6. More test on useful materials such as concrete etc in the cell phone communication would be better.
Reviewer 2 Report
1. Please explain in more details why the D-band can be a target for future 5G applications.
2. In eq. (2) please define D ( and its measurements units) and check again the equation
3. In row 185 please specify the type of LNA and give its parameters in terms of noise figure
4. Please insert o photo of the experimental setup.
5. Please give more details about the area of the obstacles and the separation of the pots in Fig. 6.
6. For a distance of 100 m the free space path loss is about 116 dB at 150 GHz. Please give an estimate for the sensitivity of your receiver
7. The values in Table 1 are unrealistic for D band. Please try to extract more accurate values by placing the TX and RX closer to obstacles but still in the far fields regions.
Round 2
Reviewer 2 Report
Please check eq.(3), the denominator exp(200D-1) is not plausible.
